# Advances of Protein Palmitoylation in Tumor Cell Deaths

**DOI:** 10.3390/cancers15235503

**Published:** 2023-11-21

**Authors:** Xiangyi Lin, Yuxuan Shi, Yuxin Zhan, Yuying Xing, Yu Li, Zhiqing Zhou, Guoan Chen

**Affiliations:** Department of Human Cell Biology and Genetics, Joint Laboratory of Guangdong-Hong Kong Universities for Vascular Homeostasis and Diseases, School of Medicine, Southern University of Science and Technology, Shenzhen 518000, China; 12013039@mail.sustech.edu.cn (X.L.); 12010139@mail.sustech.edu.cn (Y.S.); 12012741@mail.sustech.edu.cn (Y.Z.); 12010642@mail.sustech.edu.cn (Y.X.); 12131344@mail.sustech.edu.cn (Y.L.); 12031153@mail.sustech.edu.cn (Z.Z.)

**Keywords:** protein palmitoylation, apoptosis, autophagy, ferroptosis, pyroptosis

## Abstract

**Simple Summary:**

This review seeks to explore and understand palmitoylation, a crucial cellular process, and its significant role in cancer cell development and death. The research team will focus on understanding the complex relationships between palmitoylation and cancer formation, particularly how specific family proteins and enzymes contribute to, or counteract, the process. With the advent of innovative detection techniques, our understanding of these processes has grown more nuanced and sophisticated. The research will also spotlight potential agents that show promise in targeting palmitoylation for cancer therapy, offering a glimpse into potentially groundbreaking treatment strategies. Through this comprehensive review, the authors aim not only to highlight the transformative potential of studying palmitoylation in cancer treatment but also to deepen the scientific community’s understanding of the molecular mechanisms at play in cancer.

**Abstract:**

In this comprehensive survey, we delve into the multifaceted role of palmitoylation across various cell death modalities in the oncological context, from its intricate correlations with tumorigenesis, steered by the Asp-His-His-Cys tetrapeptide motif (DHHC) family, to the counter-process of depalmitoylation mediated by enzymes like Palmitoyl protein thioesterase-1 (PPT1). Innovations in detection methodologies have paralleled our growing understanding, transitioning from rudimentary techniques to sophisticated modern methods. Central to our discourse are agents like Ezurpimtrostat (GNS561) and dimeric chloroquine (DC661), promising heralds in palmitoylation-targeted cancer therapy. Collectively, this review accentuates palmitoylation’s transformative potential in oncology, foreshadowing groundbreaking therapeutic strategies and deepening our molecular comprehension of cancer dynamics.

## 1. Introduction

Post-translational modifications (PTMs) serve as molecular conductors, orchestrating protein functions throughout cellular processes, and significantly impacting health and disease. Well-established PTMs such as phosphorylation, glycosylation, and ubiquitination have traditionally captured the focus of biomedical research [1]. Nonetheless, palmitoylation has come to be recognized as an essential contributor within this field. It plays a critical role in a variety of cellular signaling pathways, including those associated with Epidermal Growth Factor Receptor (EGFR), Ras, and Programmed Cell Death Protein 1 (PD-1)/Programmed Death-Ligand 1 (PD-L1), and is vital in metabolic regulation [2,3]. In the context of cancer, the role of palmitoylation in tumorigenesis is becoming increasingly evident. An expanding array of palmitoylated proteins has been identified as oncogenic factors, with their ability to regulate protein localization, secretion, and stability. This modulation, primarily through altered affinities between proteins and cellular membranes, underscores their potential significance in the complex mechanisms of cancer [4].

Cell death, an essential biological process, plays a pivotal role in maintaining tissue homeostasis and fostering development across diverse species. This process is intricately regulated and includes distinct modes such as apoptosis, autophagy, necroptosis, ferroptosis, and pyroptosis, each with unique morphological and biochemical signatures [5,6]. In the context of oncology, aberrant cell death pathways can initiate and promote tumorigenesis and influence therapeutic outcomes [7,8]. For example, the evasion of apoptosis is a recognized cancer hallmark, conferring a proliferative advantage to malignant cells [9]. In contrast, targeting non-apoptotic cell death pathways has emerged as a therapeutic strategy for combatting apoptosis-resistant cancers [10]. The complex relationship between cell death mechanisms and cancer highlights potential therapeutic opportunities but also presents challenges to existing paradigms, thereby driving the need for innovative research approaches [11]. An enhanced comprehension of these dynamics holds the promise of improving therapeutic strategies and broadening the spectrum of interventions against cancer [12,13].

This review illuminates the increasingly recognized role of protein palmitoylation in the regulation of cell death and oncogenesis. It offers a detailed exploration of how palmitoylation modulates various cell death pathways, including apoptosis, autophagy, ferroptosis, and pyroptosis, as well as its interaction with other post-translational modifications and signaling cascades. Additionally, the review delves into the potential of palmitoylation patterns as biomarkers and their implications in the development of pharmacological interventions. By discussing the translational applications for crafting innovative cancer treatments, this review takes an important step in bridging fundamental palmitoylation research with the advancement of clinical cancer therapies.

## 2. Methods

To establish foundational understanding, the authoritative literature published post-2000 was selected for citation. In addition, to ensure a comprehensive retrieval of the literature on the role of protein palmitoylation in various forms of tumor cell death, a systematic search strategy was implemented using PubMed as the primary database and Google Scholar for supplemental searches. The PubMed search was conducted with no date restrictions to encompass the full breadth of available literature. The search terms were constructed to capture the nexus of cancer, protein palmitoylation, and the diverse modalities of cell death. The specific query string used was: “cancer” AND “palmitoylation” AND (“apoptosis” OR “autophagy” OR “ferroptosis” OR “pyroptosis”) [Title/Abstract]. To complement the PubMed search and mitigate the risk of missing relevant studies not indexed therein, a follow-up search was conducted using Google Scholar. The same search terms were applied to Google Scholar with a manual filtering process to exclude duplicates already obtained from PubMed and to select studies that met the inclusion criteria set forth for this review.

## 3. Cell Death

In the realm of cancer biology, the role of cell death transcends being simply an end-stage event, emerging as a process with critical therapeutic ramifications. This review focuses on regulated cell death (RCD) within oncological settings, where it acts as a central modulator of disease progression. RCD encompasses mechanisms like apoptosis, pyroptosis, and ferroptosis, distinguished by their contribution to immunomodulation, particularly via the release of damage-associated molecular patterns that shape anti-tumor immune responses [14,15]. Concurrently, autophagy, while typically serving as a cell-survival mechanism, can become fatal under certain pathological conditions, adding to cellular mortality. The complex interplay between autophagy and RCD mechanisms in cancer highlights the intricacy of cellular death pathways (Table 1).

Apoptosis, an intricately regulated process requiring energy, is crucial for maintaining cellular equilibrium in multicellular organisms by selectively eliminating damaged or surplus cells. This process is characterized by cellular shrinkage, nuclear fragmentation, and chromatin condensation [11]. It is governed primarily by two pathways: the intrinsic (mitochondrial) pathway and the extrinsic (death receptor) pathway [5]. The intrinsic pathway, triggered by internal stressors such as DNA damage or oxidative stress, prominently features the p53 protein, often referred to as the “genome’s guardian”. This protein initiates apoptosis to prevent the proliferation of severely damaged cells [16]. Additionally, endoplasmic reticulum (ER) stress can activate this pathway through the induction of CHOP/GADD153, in response to the accumulation of misfolded proteins or calcium imbalance. In this context, X-box binding protein 1 (XBP1) operates as a crucial transcription factor in the unfolded protein response, a mechanism essential for cellular stability under ER stress. Such a dual response highlights the elaborate cellular strategies to counterbalance stress and ensure proteostasis [17,18]. In contrast, the extrinsic apoptotic pathway, a pivotal component of programmed cell death, is initiated by extracellular death ligands that engage with death receptors on the cell surface. Notably, the Tumor Necrosis Factor Receptor (TNF-R1) and CD95 (Fas) are significant receptors that, upon ligand binding, prompt the formation of the death-inducing signaling complex (DISC). This complex triggers a sequence of events that activate initiator caspases, such as caspase-8, leading to the eventual activation of executioner caspases that dismantle the cell [19,20].

Autophagy, a conserved mechanism in eukaryotes, maintains cellular homeostasis by degrading and recycling cytoplasmic components [21]. This process is characterized by the formation of an autophagosome, a double-membraned vesicle that encapsulates cellular material for degradation and is induced by cellular stress. Following its initiation, the outer membrane of the autophagosome merges with lysosomes, facilitating the breakdown of the contained material [22]. Autophagy is regulated by the protein kinase mTOR and autophagy-related genes, components of the cellular degradation pathway [23]. The PI3K/AKT/mTOR pathway, vital for cell growth and survival, is often dysregulated in cancers [24]. Importantly, autophagy has been associated with various cell death modalities, including apoptosis and necrosis [25,26]. For instance, autophagy-related genes can mitigate the effects of anti-apoptotic proteins and promote the release of apoptogenic factors, thus increasing the propensity for apoptosis [27,28,29].

Ferroptosis is a form of iron-dependent cell death characterized by significant mitochondrial shrinkage, increased membrane density, and a decrease or loss of mitochondrial cristae [30]. Lipid peroxidation is the primary mechanism that drives ferroptosis, culminating in membrane damage [8]. This process is regulated by various oxidative and antioxidative systems. Notably, the iron-mediated Fenton reaction contributes to the accumulation of phospholipid hydroperoxides, thereby facilitating lipid peroxidation [31]. A critical component of the antioxidative defense is the system xc-, comprising the SLC7A11 and SLC3A2 subunits, which is essential for the uptake of cystine. This cystine is subsequently transformed into cysteine, a precursor for glutathione synthesis. Glutathione plays a pivotal role in the function of GPX4, an enzyme that mitigates lipid peroxidation and protects cells against ferroptosis [10,32,33,34].

Palmitoylation significantly influences chemotherapy-induced pyroptosis, a form of inflammatory cell death [35]. Pyroptosis is initiated by the activation of inflammatory caspases, which cleave Gasdermin proteins, culminating in plasma membrane disruption and cell death [36,37]. Specifically, caspase-3 activation cleaves Gasdermin E (GSDME) into its N-terminal (active) and C-terminal fragments. The active N-terminal fragment then integrates into the cell membrane, forming pores that promote ion and water influx, leading to cellular swelling, membrane rupture, and the release of inflammatory cytokines, characteristic of pyroptosis [38,39,40,41]. Furthermore, it is established that caspase-3 preferentially cleaves GSDME to initiate pore formation on the cell membrane, rather than Gasdermin D [42,43].

## 4. Palmitoylation

Palmitoylation is a multifaceted post-translational modification, encompassing S-palmitoylation, O-palmitoylation, and N-palmitoylation types. Of these, S-palmitoylation is the most common, involving a reversible covalent bond between the fatty acid palmitate and the thiol group of a cysteine residue within a protein [44]. This form of palmitoylation is dynamic, with its temporal oscillations ranging from mere seconds to several hours, depending on cellular signaling [4]. On the other hand, O-palmitoylation and N-palmitoylation refer to the addition of fatty acyl groups to serine residues and the protein’s amino terminus, respectively, as detailed in Table 2 and illustrated in Figure 1.

The S-palmitoylation process relies fundamentally on the catalytic capabilities of palmitoyl S-acyltransferases (PATs), distinguished by the conserved DHHC zinc-finger domain (ZDHHC)—a hallmark motif across the DHHC protein family. The number of DHHC family members varies significantly, from seven in Saccharomyces cerevisiae to twenty-three in mammals, as noted in reference [45]. Gene expression patterns within this family demonstrate notable associations with cancer development. The work of Ko and Dixon provides a detailed account of the correlation between DHHC gene expressions and various cancers up to 2018 [46]. Recent studies indicate subtle changes in DHHC expression in several common cancers over the subsequent five years, as detailed in Table 3. For instance, ZDHHC1 shows decreased expression in breast, prostate, and gastric cancers but is overexpressed in endometrial, renal, and pancreatic cancers. ZDHHC14 displays an altered expression in prostate and testicular germ cell tumors, while ZDHHC20’s expression rises across various cancers, including ovarian, breast, and colon. Notably, ZDHHC15 and ZDHHC16 are primarily reduced in brain cancers, particularly glioblastomas. Additionally, the overexpression of ZDHHC9 is observed in a range of cancers from breast to glioblastoma. These patterns highlight the potential role of DHHC family members as biomarkers for diagnosis or as targets for cancer therapy. More research is needed to clarify their specific roles in the molecular mechanisms of cancer progression and pathogenesis.

Within the complex arena of cellular protein modifications, depalmitoylation stands out as a crucial process in the strategic removal of palmitoyl groups [66]. At the forefront of this process are three key enzymes: Palmitoyl protein thioesterase-1 (PPT1), Acyl protein thioesterase (APT), and Abhydrolase domain-containing protein 17 (ABHD17). PPT1, a distinguished member of the lipoacyl thioesterase family, expertly reverts proteins to their original state, with its dysfunction being associated with neurodegenerative disorders such as neuronal ceroid lipofuscinosis [67,68]. APT skillfully modifies the palmitoylation profile of essential proteins like PSD-95, influencing synaptic strength and cognitive functions significantly [69]. ABHD17, notably, has garnered attention in oncology for its capacity to alter NRAS-driven cancer dynamics, presenting itself as an appealing target for therapeutic development [70,71] (Table 4).

Over the past fifty years, the techniques for detecting palmitoylation—a crucial post-translational modification—have been significantly refined. Initially, in the 1970s, researchers labeled proteins with [3H]-Palmitate, facilitating their detection via autoradiography after SDS-PAGE. In subsequent decades, the application of [125I]-Iodopalmitate for metabolic labeling enhanced the specificity of identifying palmitoylated proteins [72]. Entering the 21st century, mass spectrometry became essential, offering precise identification of palmitoylated proteins and their modification sites [72,73,74].

The 2000s introduced the Acyl-Biotin Exchange (ABE) method, which leveraged the hydroxylamine-induced cleavage of palmitate followed by biotinylation of the resultant cysteines [72,75]. The 2010s saw a further innovation with the Acyl-Resin Assisted Capture (Acyl-RAC) technique, which simplified the ABE process by directly capturing depalmitoylated proteins onto a thiol-reactive resin [72,76]. In parallel, Click Chemistry’s advent, using bioorthogonal reactions with fatty acids, expanded the scope of research [72,73,77]. Additionally, the decade introduced the Proximity Ligation Assay (PLA) for a novel in situ detection, relying on antibody pairing for signal amplification [78,79]. The period also marked the debut of PalmPISC, an integrated method combining metabolic labeling, click chemistry, and mass spectrometry [80,81]. These developments highlight the evolving landscape of palmitoylation research and its essential role in cellular biochemistry, as summarized in Table 5.

## 5. Palmitoylation in Tumor Apoptosis

In the intrinsic apoptosis pathway, the tumor suppressor p53 plays a crucial role in halting the cell cycle at the G1/S checkpoint through selective DNA binding in the late G1 phase [82]. Decreased expression of ZDHHC16 in glioblastomas, commonly characterized by EGFR amplification, leads to p53 activation and subsequent cell cycle arrest at the G1/S checkpoint [60,61]. Additionally, p53 promotes gliomagenesis by enhancing the self-renewal and tumorigenic potential of glioma stem-like cells. This is achieved through the upregulation of ZDHHC5 transcription and the modification of EZH2 palmitoylation [52]. ZDHHC5-mediated palmitoylation of EZH2 activates histone H3 Lysine 27 trimethylation (H3K27me3), suppresses miR-1275, elevates Glial Fibrillary Acidic Protein (GFAP) expression, and decreases DNA methyltransferase 3 alpha (DNMT3A) binding to the OCT4 promoter, contributing to glioma malignancy [83]. Furthermore, p53 is palmitoylated by ZDHHC1, which influences DNMT3A’s association with the ZDHHC1 promoter, leading to its hypermethylation and the subsequent inactivation and degradation of non-palmitoylated p53, thus facilitating tumorigenesis [48]. Meanwhile, the SUMOylation of XBP1 is compromised by reduced palmitoylation, diminishing its transcriptional activity and underscoring the role of post-translational modifications in tumor progression [84]. (Figure 1). Contrastingly, the S-palmitoylation of Bcl-2-associated X protein (BAX) at Cys-126, essential for initiating apoptosis, illustrates a divergence in response between normal and cancerous cells, with the former demonstrating a more rapid apoptotic response due to increased BAX S-palmitoylation [85].

The extrinsic apoptotic pathway is stringently modulated by protein modifications, including palmitoylation. Palmitoylation at cysteine 199 in humans and cysteine 194 in mice is critical for CD95 function, facilitating its clustering, formation of microaggregates, and internalization, which are essential for caspase-8 activation and the induction of apoptosis [86]. Upon palmitoylation, CD95 moves to lipid rafts enriched with cholesterol and sphingolipids, which act as pivotal signaling platforms [87]. Nonetheless, this apoptotic mechanism is disrupted in certain malignancies, such as Chronic Lymphocytic Leukemia, where the upregulation of APTs-governed by microRNAs like miR-138 and miR-424-disrupts CD95 palmitoylation, hinders apoptosis, and contributes to treatment resistance [88]. Moreover, the functionality of the extrinsic pathway is also affected by the regulation of TNF-R1, another key death receptor. The inhibition of APT2 causes an increase in TNF-R1 palmitoylation and its stabilization on the cell surface, which decreases internalization and downstream apoptotic signaling. Conversely, APT2 knockdown boosts TNF-R1 surface presence but deters apoptosis, a resistance analogous to that observed in cancers where CD95 palmitoylation is suppressed due to APT overexpression [89]. In the realm of tumor suppression, p53 is integral; it upregulates RARRES3 expression, which in turn impedes the Wnt/β-catenin signaling pathway through the deacylation of its components [90]. This RARRES3 activity inhibits cancer cell traits like proliferation, epithelial-mesenchymal transition (EMT), and stemness in breast cancer, making the dynamic between RARRES3, p53, and Wnt/β-catenin signaling a significant focus for therapeutic strategies in oncology, revealing new insights into oncogenic mechanisms [91] (Figure 2).

## 6. Palmitoylation in Tumor Autophagy

Autophagy is frequently utilized as a mechanism to sustain cellular viability; however, it can also be associated with cell death [41]. It has been established that autophagy-mediated cell death and apoptosis share complex interrelationships. At the heart of this research is the AKT protein, which is crucial in regulating various cellular processes. Dysregulation of AKT has been implicated in a range of pathological conditions, particularly tumor progression due to AKT hyperactivation [92]. Among post-translational modifications, S-palmitoylation at the conserved residue C344 in hAKT1 is of particular interest. Alterations at this site, such as those observed in the AKT1-C344S mutant, affect AKT signaling and demonstrate a significant connection between AKT’s S-palmitoylation and cellular outcomes. Notably, this mutation alters lysosomal interactions during autophagy and reduces adipogenic differentiation potential in vitro, as depicted in Figure 2 [93].

Concurrently, the lysosomal enzyme PPT1, which cleaves thioester bonds in S-acylated proteins, presents intriguing consequences [67]. Elevated PPT1 levels are associated with increased AKT phosphorylation and enhanced cellular resistance to apoptotic stimuli. An analysis of hepatocellular carcinoma (HCC) using the TCGA dataset highlights the significance of lysosomal genes, with PPT1 positioned as a key player. By elucidating the role of PPT1 in depalmitoylation, it holds the potential to pioneer new therapeutic strategies for HCC [94].

In the dynamic field of oncology, the therapeutic value of PPT1 inhibitors, such as GNS561 and dimeric chloroquine (DC661), is being recognized [95,96]. DC661 surpasses hydroxychloroquine in inhibiting autophagy by targeting PPT1, while GNS561 shows potential synergy with immune checkpoint inhibitors, marking a shift in cancer treatment strategies [68,97]. Furthermore, the Phase 1b trial of GNS561 reveals its unique mechanism involving lysosomal Zn(2+) accumulation, underlining its therapeutic promise [98].

The influence of PPT1 extends beyond HCC. For example, in melanoma, inhibiting PPT1 enhances the efficacy of anti-PD-1 antibodies, potentially boosting T-cell mediated tumor destruction and revolutionizing melanoma therapies [99]. The interaction between ferroptosis and palmitoylation presents viable strategies for cancer treatment. In colorectal cancer, palmitoylation-driven lysosomal degradation of Interferon gamma receptor 1 (IFNGR1), upon optineurin depletion, impairs the IFNγ and MHC-I pathways, leading to immune evasion and intrinsic resistance to immunotherapy. Interestingly, manipulating IFNGR1 palmitoylation may intensify T-cell immune responses and improve the effectiveness of checkpoint inhibitors [100,101]. A significant finding is that the palmitoylation of the cytoplasmic domain of PD-L1 prevents its lysosomal degradation. The inhibition of this palmitoylation, or the responsible acyltransferase ZDHHC3, may strengthen antitumor immunity, offering new strategies to combat the PD-L1-mediated immune escape in cancer [102].

The novel compound DQ661, a dimeric quinacrine derivative, highlights the crucial role of PPT1 in lysosomal function, with implications for cancer and therapeutic resistance [103]. Targeted inhibition of PPT1 disrupts both the mTOR pathway and lysosomal function, signaling a new direction in cancer therapy [104]. Furthermore, the interaction among PPT1, TORC1, and HSP90 is becoming increasingly significant in the context of treatment resistance, indicating a potentially synergistic approach using mTOR and HSP90 inhibitors [105].

Inhibiting protein depalmitoylation promotes tumor cell death. A synthetic analog, DAPKA, has been shown to inhibit PPT1 in a specific fluorescence-based assay and to enhance the cytotoxicity of chemotherapeutic agents like etoposide and adriamycin in neurotumor cell lines. Overexpression of PPT1 conferred resistance to apoptosis, suggesting that its inhibition could potentiate the effects of these chemotherapeutic agents [106] (Figure 3).

In conclusion, the interconnection between palmitoylation and autophagy, reinforced by novel insights into PPT1, offers a hopeful prospect for innovative cancer treatments [106] (Figure 3).

## 7. Palmitoylation in Ferroptosis and Pyroptosis

Ferroptosis is a type of cell death characterized by the accumulation of lipid peroxides, with SLC7A11 playing a crucial role in its regulation. This mechanism is particularly relevant in the oncological setting; for instance, in hepatocellular carcinoma, ferroptosis resistance—mediated by lncRNA DUXAP8 and SLC7A11—diminishes the efficacy of treatments such as sorafenib, highlighting the need for novel therapeutic strategies [107,108]. Moreover, complex interactions exist among various cell death pathways, including pyroptosis, ferroptosis, and antitumor immunity. CD8+ T cells, for example, release Granzyme A, which cleaves Gasdermin B and triggers pyroptosis. Concurrently, IFN-γ produced by CD8+ T cells suppresses SLC7A11, leading to lipid ROS accumulation and ferroptosis induction. Palmitoyltransferases ZDHHC 2, 7, 11, and 15 modify GSDME-C at Cys407 and Cys408, facilitating its separation from GSDME-N and enhancing pyroptosis [109]. The suppression of ZDHHC1 through promoter methylation may increase oxidative and ER stress, promoting pyroptosis via the potential substrate NLRP3, thus implicating ZDHHC1 in tumor-suppressive mechanisms [47]. Furthermore, erianin, noted for its anticancer effects against oral squamous cell carcinoma, inhibits cell proliferation both in vitro and in vivo. It induces a G2/M phase arrest, apoptosis, and a novel form of GSDME-mediated pyroptosis, while also affecting autophagy [110] (Figure 4).

## 8. Palmitoylation in Oncology: A Multifaceted Therapeutic Target

In the complex field of oncology, palmitoylation has emerged as a critical therapeutic target, revealing new pathways in cellular processes previously not well understood (Table 6). Palmitoylation, particularly when mediated by DHHC3, inhibits PD-L1 ubiquitination and prevents its association with the ESCRT machinery, thus blocking its internalization into multivesicular bodies (MVBs) and lysosomes. Pharmacological interventions, such as with 2-Bromopalmitate (2-BP) or competitive inhibitory peptides, have been shown to increase PD-L1 ubiquitination and its subsequent degradation. This process enhances the cytotoxicity of tumor-specific T-cells both in vitro and in vivo, as evidenced by studies involving 2-BP [102,111].

Advancing these developments, compounds like GNS561 and DC661 represent innovative strategies for modulating lysosomal functions and autophagy by targeting the PPT1 pathway. Preliminary clinical trials of GNS561 have indicated encouraging results, albeit not definitive results [96,98]. The PPT1 inhibitor DC661 has notably improved the anti-tumor immune response. To reduce the side effects associated with DC661, there is ongoing development of advanced organic nanocarriers that possess self-regulating features, aiming to enhance the precision of tumor-targeting drug delivery and therapeutic efficacy [112].

In melanoma treatment, PPT1 inhibition significantly enhances the efficacy of anti-PD-1 antibodies, marking a substantial advancement in therapeutic strategies. The hypothesized mechanism involves bolstering tumor immunity by inducing macrophage polarization from M2 to M1 phenotype, reducing myeloid-derived suppressor cell populations within the tumor environment, and stimulating macrophages to release interferon-beta, which in turn promotes T-cell-mediated cytotoxicity. Contrarily, hydroxychloroquine has not demonstrated antitumor activity as a monotherapy in melanoma models, aligning with clinical observations [99].

Furthermore, PPT1’s role is implicated in various malignancies. For instance, erianin impedes growth by downregulating PPT1 in Oral Squamous Carcinoma Cells and provokes G2/M-phase arrest, apoptosis, and autophagy via the ROS/JNK Signaling Pathway in Human Osteosarcoma Cells, as corroborated by both in vitro and in vivo studies [113,114]. In chronic lymphocytic leukemia, palmitoylation’s significance is underscored through its involvement with CD95 and the inhibition of N-Ras depalmitoylation by ABD957 [70,71].

Recent research also explores the characteristics of N-palmitoylated peptides P1MK5E and P1MEK5. These peptides undergo pH-sensitive structural alterations; particularly, P1MK5E exhibits increased anti-cancer activity in A549 cells attributed to its α-helical structure [115]. Additionally, Artonin F’s inhibition of c-MET palmitoylation in lung cancer introduces potential new therapeutic pathways by modulating stability and signaling [116] (Table 6).

**Table 6 cancers-15-05503-t006:** Therapeutic Implications and Strategies in Oncology.

Target/Pathway	Agent/Influence	Clinical Potential and Remarks	Reference
A general inhibitor of protein S-palmitoylation	2-BP	Inhibit S-palmitoylation: e.g., Inhibit DHHC3 acyltransferase as a promising therapeutic avenue towards enhancing tumor-specific immunity	[102]
Pancreatic cancer	2-BP and PD-1/PD-L1	Facilitate checkpoint immunotherapy	[111]
HCC	GNS561	Inhibit PPT1 to inhibit autophagy	[98]
HCC	DC661	Inhibit PPT1 to inhibit autophagy	[112]
Melanoma	DC661 and anti-PD-1 antibody	Synergistic enhancement of antitumor activity	[99]
Melanoma, pancreatic cancer, and colorectal cancer	DQ661	Inhibit PPT1 to inhibit autophagy and mTORC1 activity by specifically targeting PPT1.	[103]
Oral squamous cell carcinoma, osteosarcoma, hepatic cancer, lung cancer, and cervical cancer	Erianin	Promote apoptosis, autophagy, ferroptosis, and pyroptosis and enhance the immunotherapy	[110]
Chronic lymphocytic leukemia	ABD957	Targeted disruption of N-Ras depalmitoylation; Synergy with MEK inhibition	[70,71]
Lung Cancer	P1MK5E	Enhance Necroptosis and Anticancer activity	[115]
Lung Cancer	Artonin F	c-Met in Competitive inhibition of c-Met palmitoylation	[116]

## 9. Future Perspective

### 9.1. Current Understanding and Challenges in Protein Palmitoylation

Current understanding of protein palmitoylation underscores its complex integration within cellular processes, orchestrated primarily by the ZDHHC enzyme family. Yet, the specificity of palmitoylation, which varies according to the cellular proteome, presents a considerable challenge: devising a systematic approach to map these events across different cell types and contexts [117,118,119]. Despite advancements in technologies like acyl-biotin exchange and mass spectrometry, which have improved our ability to delineate these modifications, they still face inherent limitations [120]. These hurdles impede a comprehensive understanding of the interplay between palmitoylation, the cellular environment, and the diversity of protein targets.

### 9.2. Linking Palmitoylation to Cellular Death in Cancer

Within the domain of oncology, palmitoylation has been recognized as a pivotal modulator of cell death mechanisms. Altered patterns of palmitoylation can undermine the normal apoptotic pathways, enabling cancer cells to circumvent programmed cell death and sustain survival [46]. The etiology of these disruptions includes changes in membrane dynamics and protein localization, which are critical in cellular signaling and initiating apoptosis. Additionally, anomalies in ZDHHC enzyme function have been associated with the proliferation of cancer characteristics, including unrestrained cell growth, invasion, and metastasis [121]. These findings underscore the enzymes’ dual capacities as potential oncogenes or tumor suppressors, contingent on the biological context, and spotlight the therapeutic promise of targeting palmitoylation pathways to restore regulated cell death and combat cancer progression [111,122].

### 9.3. Therapeutic Implications and Future Directions

The landscape of cancer therapy stands on the brink of a transformative shift with the emerging focus on palmitoylation pathways. The dynamic ‘palmitome’—an intricate network within cellular systems—opens new avenues for precision oncology interventions. However, the path toward targeted therapies is fraught with challenges, including off-target effects, the complexity of palmitoylation dynamics, and potential impacts on normal cellular functions and immune responses [122]. In response, our strategy must involve meticulous target selection, dosing regimens, and the identification of biomarkers to inform personalized treatments [123]. Advances in molecular insights and technological breakthroughs promise to pave novel therapeutic avenues, potentially redefining cancer metabolism and introducing innovative strategies to combat tumor proliferation and resistance mechanisms. Considering these advancements, the targeted modulation of palmitoylation pathways could herald a new era in precision oncology [124].

## 10. Conclusions

Our expanding knowledge of PTMs, particularly palmitoylation, underscores their critical roles in cellular operations and disease states. Palmitoylation plays a key role in modulating proteins, directing cellular pathways such as apoptosis, autophagy, ferroptosis, and pyroptosis. These pathways significantly influence chemotherapy sensitivity, neurodegenerative diseases, and cell death mechanisms.

As we delve deeper into PTMs, especially palmitoylation, their intricate roles across various cellular contexts become increasingly apparent, offering a wealth of insights. Understanding the complex interplay of concurrent modifications at identical cellular sites may hold the key to unraveling the extensive panorama of tumor cell death. Indisputably, as we unveil the profound complexities of PTMs, we approach a revolutionary juncture with the potential to redefine the direction of oncological research and therapy.

## Figures and Tables

**Figure 1 cancers-15-05503-f001:**
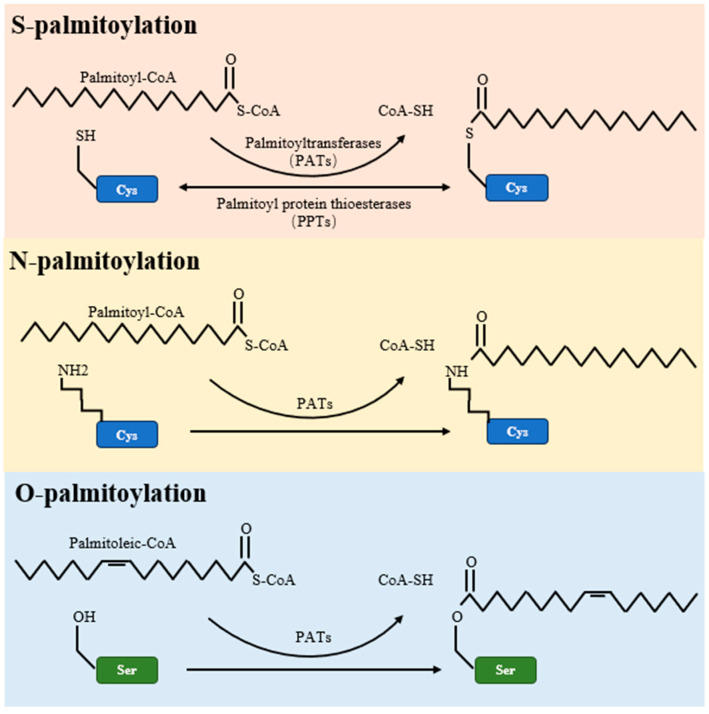
Schematic representations of S-palmitoylation, O-palmitoylation, and N-palmitoylation reactions. The figure elucidates the enzymatic mechanisms underlying protein palmitoylation. Palmitoyltransferases (PATs) mediate the transfer of palmitate from palmitoyl-CoA to target proteins, marking S-palmitoylation as typically reversible. Conversely, O-palmitoylation and N-palmitoylation are shown as irreversible alterations. The figure further identifies PPTs as the enzymes that reverse S-palmitoylation, thereby enabling the dynamic control of this post-translational modification.

**Figure 2 cancers-15-05503-f002:**
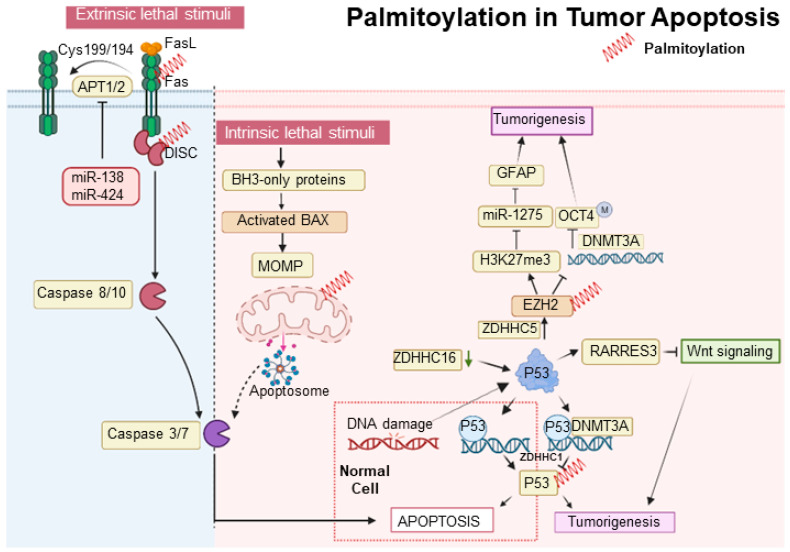
Palmitoylation in tumor apoptosis and cancer progression. The palmitoylation modification of proteins such as p53, BAX, and CD95 (Fas) plays a pivotal role in modulating apoptosis and influencing tumor progression. Proteins that undergo palmitoylation are denoted by a red spring.

**Figure 3 cancers-15-05503-f003:**
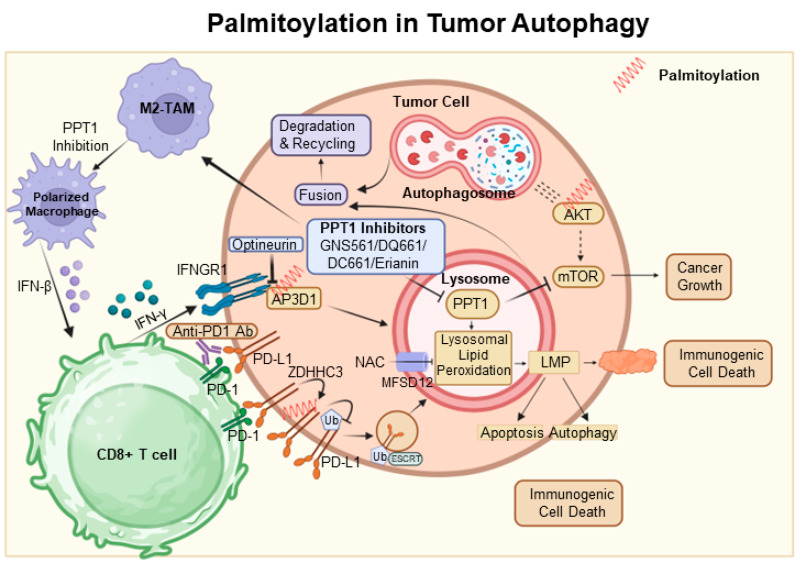
Palmitoylation in autophagic signaling and cancer progression. It highlights the growing importance of PPT1 in the regulation of apoptosis, autophagy, and AKT phosphorylation. Moreover, it points to the therapeutic potential of PPT1 inhibitors, such as GNS561 and DC661, in treating various cancers. The intricate relationship between PPT1 and key cellular pathways opens up new prospects for innovative cancer therapies. In this context, ZDHHC3 functions as a palmitoyltransferase, while PPT1 acts as a depalmitoylating enzyme. Proteins undergoing palmitoylation, including AKT, IFNGR1, and PD-L1, are denoted by red springs.

**Figure 4 cancers-15-05503-f004:**
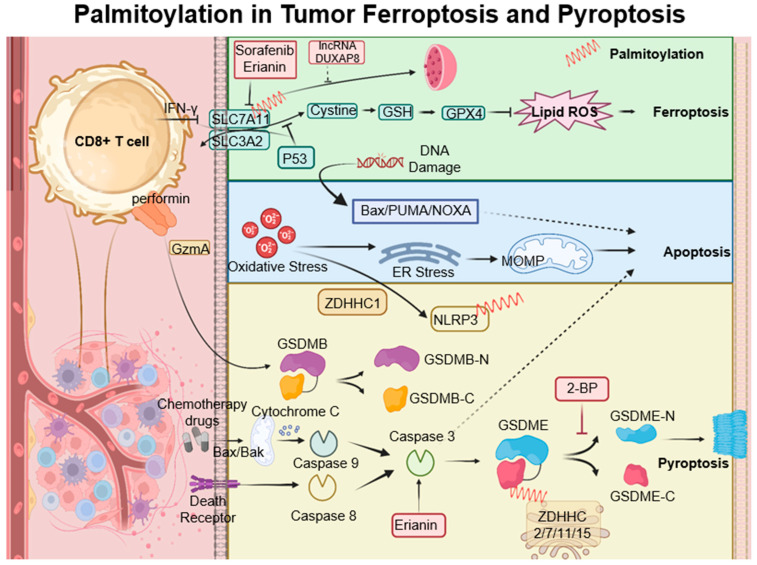
Palmitoylation-modulated interplay of ferroptosis and pyroptosis in oncological contexts. Ferroptosis is closely associated with critical molecules such as SLC7A11, which affect glutathione synthesis and the activity of GPX4. Palmitoylation plays a crucial role in chemically induced pyroptosis by regulating the cleavage of Gasdermin proteins, leading to inflammatory responses. CD8+ T cells influence both pyroptosis and ferroptosis via the secretion of GzmA protein and the action of IFN-γ, respectively. This sophisticated interplay between cell death pathways and immune responses opens up potential strategies for novel cancer treatments, as demonstrated by the use of therapeutic agents like erianin. ZDHHCs, which are palmitoyltransferases, modify proteins such as SLC7A11, NLRP3, and GSDME through palmitoylation, depicted as red springs.

**Table 1 cancers-15-05503-t001:** Comparisons of different types of cell deaths.

Palmitoylation-Related Cell Death	Extrinsic Apoptosis	Intrinsic Apoptosis		Autophagy	Ferroptosis	Pyroptosis
Morphological features	Cellular contraction, nuclear fragmentation, and chromatin condensation.			Double-membrane autophagosome merging with lysosome	Reduction in mitochondrial volume, increase in mitochondrial membrane density, disappearance of mitochondrial cristae	Cells continue to swell until the cell membrane ruptures, leading to a massive release of cell contents and pro-inflammatory factors
Activation Signals	Initiated when death ligands bind to cell surface receptors	DNA damage, oxidative stress	ER stress from misfolded proteins or calcium imbalance	Cellular stress	Oxidative stress	Microbial infections and non-infectious stimuli
Key Proteins and Molecules	Death-inducing signaling complex (DISC) and Initiator caspases	p53 protein	CHOP/GADD153	Akt, mTOR and PI3K	GPX4 and system xc-	Gasdermin and caspase

**Table 2 cancers-15-05503-t002:** Overview of Protein Palmitoylation Types.

Palmitoylation Type	S-Palmitoylation	O-Palmitoylation	N-Palmitoylation
Amino Acid Residue Involved	Cysteine (Cys)	Serine (Ser) or Threonine (Thr)	N-terminal amino group
Bond Type	Thioester	Ester	Amide
Reversibility	Reversible	Typically irreversible	Typically irreversible
Enzymes Involved	Palmitoylltransferases (PATs) for addition, Acyl-protein thioesterases (APTs) or Palmitoyl-protein thioesterase (PPTs) for removal	Not well-defined	N-myristoyltransferase (NMT), though this is for myristoylation which is more common at the N-terminus
Function	Regulation of protein-membrane association, protein–protein interactions, and protein stability	Rare, less studied; potential role in regulation of protein stability and function	Rare; N-terminal myristoylation is more common, plays role in protein-membrane association and stability
Examples of Proteins	Ras proteins, G-protein α subunits, PSD-95	Not well characterized due to its rarity	Rare, more common is N-myristoylation (e.g., Src kinase)
Pathways/Processes Involved	Signal transduction, apoptosis, synaptic plasticity	Not well-defined due to rarity	Not well-defined due to rarity, but N-myristoylation is involved in signal transduction
Disease Associations	Neurodegenerative diseases, cancers, viral infections	Not well-characterized due to rarity	Not well-characterized due to rarity

**Table 3 cancers-15-05503-t003:** Expression alterations of DHHC family members in cancers identified within the last 5 years.

ZDHHC Family	Alteration	Cancer	Reference
ZDHHC1	Downregulated	Breast, prostate, and gastric cancers	[47,48]
	Upregulated	Endometrial, renal, and pancreatic cancers	[46]
ZDHHC2	Downregulated	Pancreatic cancer	[3]
	Upregulated	Renal cancer	[46]
ZDHHC3	Upregulated	Breast, prostate, renal, and colorectal cancers	[49,50]
ZDHHC4	Upregulated	Renal cancer	[46]
ZDHHC5	Upregulated	Lung adenocarcinoma, glioma, and breast cancer,	[51,52]
ZDHHC7	Downregulated	Colorectal Cancer	[3]
ZDHHC9	Upregulated	Breast, colorectal, myeloma, glioblastoma, and prostate cancer	[53]
ZDHHC11	Upregulated	Burkitt lymphoma	[54]
ZDHHC11B	Downregulated	Lung adenocarcinoma	[55]
ZDHHC12	Upregulated	Glioma and ovarian cancer	[56]
ZDHHC13	Downregulated	Melanoma	[3]
ZDHHC14	Downregulated	Prostate and testicular germ cell tumor	[57]
	Upregulated	Pancreatic cancer	[46]
ZDHHC15	Downregulated	Glioblastoma, kidney renal clear cell carcinoma	[58,59]
ZDHHC16	Downregulated	Glioblastoma	[60,61]
ZDHHC18	Upregulated	Ovarian cancer	[3]
ZDHHC19	Upregulated	Glioblastoma, cervical cancer, kidney renal clear cell carcinoma	[59,62]
ZDHHC20	Upregulated	Ovarian, breast, kidney, colon, and prostate cancer	[63]
ZDHHC21	Upregulated	Urothelial, renal, and non-small cell lung cancer	[3]
ZDHHC22	Downregulated	Estrogen receptor negative breast cancer	[64]
ZDHHC23	Upregulated	B-precursor acute lymphoblastic leukemia and renal cancer	[65]

**Table 4 cancers-15-05503-t004:** Summary of depalmitoylated enzymes.

Enzyme	Function	Relevance in Tumors	Notes
PPT1	Responsible for removing palmitoyl groups from proteins	Associated with neurodegenerative diseases, but direct role in tumors remains less clear	PPT1’s primary function is to maintain stability of membrane proteins
APT	Catalyzes depalmitoylation processes, such as depalmitoylation of PSD-95	Linked to synaptic plasticity and memory formation, potentially affecting tumor-associated signaling and proliferation	Inhibition of APT1 is considered to enhance synaptic function, but its exact role in tumors requires further investigation
ABHD17	Involved in the depalmitoylation process	Viewed as a therapeutic target for NRAS mutant tumors due to its influence on the palmitoylation cycle of N-Ras	Inhibition of ABHD17 may have therapeutic potential in NRAS-driven tumors

**Table 5 cancers-15-05503-t005:** Evolution of Palmitoylation Detection Methods.

Decade	Method Name	Brief Description
1970s	Radioactive Labeling with [^3^H]-Palmitate	Proteins labeled with radioactive palmitate, detected via autoradiography post SDS-PAGE.
1980s–1990s	[^125^I]-Iodopalmitate Metabolic Labeling	Enhanced specificity through metabolic labeling with radioactive iodopalmitate.
1990s–present	Mass Spectrometry	Accurate identification of palmitoylated proteins and specific modification sites.
2000s	Acyl-Biotin Exchange	Palmitate cleavage by hydroxylamine, with biotin tagging of revealed cysteines.
2010s	Acyl-Resin Assisted Capture	Direct capture of de-palmitoylated proteins using thiol-reactive resin.
2010s	Click Chemistry	Bioorthogonal reactions with specialized fatty acids to affix reporter molecules.
2010s	Proximity Ligation Assay	In situ detection through paired antibodies, leading to oligonucleotide ligation and amplified signal.
2010s	PalmPISC	Integration of metabolic labeling, click chemistry, and mass spectrometry for comprehensive analysis.

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
