# Peer review of "Advances of Protein Palmitoylation in Tumor Cell Deaths"

_cancers, 2023, doi:10.3390/cancers15235503_

Round 1

Reviewer 1 Report

Comments and Suggestions for Authors

The present paper looks fine. I don't see any reason not to publish as it is.

Author Response

Thanks for your professional review work on our article!

Reviewer 2 Report

Comments and Suggestions for Authors

In the review manuscript, Lin et al. summarized the research progress on the roles of protein palmitoylation in cancer cell death. The review is well organized, with high-quality tables and figures. I have some suggestions for the authors to improve the manuscript.

[1] Cancer progression involves disease initiation, cancer cell proliferation, cancer cell survival and death,  tumor growth, and metastasis. The manuscript does not provide the rationale for focusing on cell death.

[2] Since this manuscript focuses on cancer cell death, a section (after Introduction and before Palmitoylation) should be added to describe different types of cell death and their definitions and comparisons. A new table (and a graphic figure) can be added.

[3] Individual types of cell death (e.g., autophagy-induced cell death) should be defined more clearly with more details; this can be updated in the new section mentioned above.

[4] An “unanswered questions” section can be added before the last section. More questions/changes, like palmitoylation’s roles in the crosstalk among different types of cell death, can be added.

[5 For English writing, the rules of scientific writing should be followed.

For example, abbreviations should be defined when first used in the abstract and the main text (e.g., “DHHC” should be defined in the abstract and the main text (defined twice)) and constantly used afterward. The authors provide definitions of abbreviations in the abbreviation list; most abbreviations are not defined in the manuscript. The manuscript uses too many words “unfamiliar to readers,” which might encourage the readers to read or finish reading the manuscript. Some “emotional” words might not benefit the accurate understanding of the main findings of the manuscript.

Comments on the Quality of English Language

For English writing, the rules of scientific writing should be followed.

For example, abbreviations should be defined when first used in the abstract and the main text (e.g., “DHHC” should be defined in the abstract and the main text (defined twice)) and constantly used afterward. The authors provide definitions of abbreviations in the abbreviation list; most abbreviations are not defined in the manuscript. The manuscript uses too many words “unfamiliar to readers,” which might encourage the readers to read or finish reading the manuscript. Some “emotional” words might not benefit the accurate understanding of the main findings of the manuscript.

Author Response

Thank you very much for taking the time to review this manuscript. Please find the detailed responses below and the corresponding revisions highlighted changes in the re-submitted files.

Comments 1: Cancer progression involves disease initiation, cancer cell proliferation, cancer cell survival and death, tumor growth, and metastasis. The manuscript does not provide the rationale for focusing on cell death.

Response 1:

Thank you for your suggestion. In the revised manuscript, we have enhanced our discussion of cell death by dedicating a distinct section to this critical topic. The importance of tumor cell death is now highlighted at the outset of the "Cell Death" section to underscore its relevance to the review,which can be found in the revised manuscript in Page 2.

Furthermore, we have refined our examination of palmitoylation, ensuring that we concentrate on the most pivotal principles related to cell death. This includes a focused discussion on key proteins and molecules that undergo palmitoylation and play a significant role in cell death pathways. As suggested, extraneous processes not critically pertinent to the scope of this review have been omitted to maintain a clear and coherent narrative.

Comments 2: Since this manuscript focuses on cancer cell death, a section (after Introduction and before Palmitoylation) should be added to describe different types of cell death and their definitions and comparisons. A new table (and a graphic figure) can be added.

Response 2: Thank you for your suggestion. We have completed the section “Cell Death” after Introduction and before Palmitoylation and deleted the corresponding description in the latter sections (including Apoptosis, Autophagy, Ferroptosis and Pyroptosis). Furthermore, we have added a new table “Comparisons of different types of cell deaths” according to the morphological features, activation signals and key proteins/molecules, which can be found in  Page 2 of the revised manuscript.

Comments 3: Individual types of cell death (e.g., autophagy-induced cell death) should be defined more clearly with more details; this can be updated in the new section mentioned above.

Response 3:

We think this is an excellent suggestion, so we take a close look on the description of each type of cell death. The details of the improvement are listed as below and can be found in Page 2 in the revised manuscript:

â‘ We have incorporated additional information regarding XBP1 in relation to its role in apoptosis, particularly noting its significance in the context of palmitoylation. Furthermore, we have enriched the manuscript with a more detailed account of death receptors, specifically highlighting the Tumor Necrosis Factor Receptor (TNF-R1) and CD95 (Fas), to provide a clearer understanding of their involvement in apoptotic processes.

â‘¡Our revised text now offers an expanded overview of autophagy, emphasizing the mechanisms of degradation and recycling of cytoplasmic components. Moreover, we have provided a focused elaboration on the involvement of mTOR signaling and autophagy-related genes, thereby giving the reader a more comprehensive insight into this cellular pathway.

â‘¢In ferroptosis, we have improved the description about lipid ROS and the system xc-, thus addressing the intricate balance of lipid peroxidation and antioxidant defenses in this form of cell death.

â‘£In pyroptosis, we have added the details about caspase-3/GSDME mediated pathway, since it is mentioned in the palmitoylation related to pyroptosis afterwards.

Comments 4: An “unanswered questions” section can be added before the last section. More questions/changes, like palmitoylation’s roles in the crosstalk among different types of cell death, can be added.

Response 4:

Thank you for your suggestion. We have introduced a new section “Future perspective” titled "Future Perspective" preceding the concluding part of our manuscript, which can be found in Page 15 in revised manuscript. This section is structured into three key subsections as follows:

① "Current Understanding and Challenges in Protein Palmitoylation," where we discuss the present knowledge landscape and pinpoint the primary challenges ahead.

② "Linking Palmitoylation to Cellular Death in Cancer," which underscores the role of palmitoylation in cancer cell fate and its impact on disease progression.

â‘¢  "Therapeutic Implications and Future Directions," where we explore potential therapeutic strategies targeting palmitoylation pathways and forecast emerging research trends.

Comments 5: For English writing, the rules of scientific writing should be followed.

For example, abbreviations should be defined when first used in the abstract and the main text (e.g., “DHHC” should be defined in the abstract and the main text (defined twice)) and constantly used afterward. The authors provide definitions of abbreviations in the abbreviation list; most abbreviations are not defined in the manuscript. The manuscript uses too many words “unfamiliar to readers,” which might encourage the readers to read or finish reading the manuscript. Some “emotional” words might not benefit the accurate understanding of the main findings of the manuscript.

Response 5: Thank you for your suggestion. We have tried our best to improve the language of the manuscript, which is highlighted in the revised manuscript. These changes will not influence the content and framework of the paper. Hopefully, the revised manuscript could be acceptable for you.

4. Response to Comments on the Quality of English Language

Response: Thank you for your suggestion. We have tried our best to improve the language of the manuscript, which is highlighted in the revised manuscript. These changes will not influence the content and framework of the paper. Hopefully, the revised manuscript could be acceptable for you.

5. Additional clarifications

We feel great thanks for your valuable feedback. In addition to the above improvement, we have further inspected and updated the literatures, ensuring the novelty of the article. Furthermore, we provided a scheme for the process of palmitoylation, making it easier for the readers to understand. Last but not least, we further checked the other errors in our article including the typographical errors.

Reviewer 3 Report

Comments and Suggestions for Authors

The manuscript by Lin et al. summaries the recent advances in the studies of protein palmitoylation in tumor cell deaths. This review is timely, well written, and easy to follow. Nevertheless, I have several suggestions that I hope will help the authors improve the manuscript.

- I would like to suggest that authors provide some specifications or more detailed descriptions for the abbreviations used in the abstract;

- In my opinion, it would be nice to add chemical schemes for S-palmitoylation, O-palmitoylation, and N-palmitoylation reactions within an additional figure;

- I think it would be helpful to the reader if the legend for the red spring representing palmitoylation will be included in the figure.

- Ref. 6 seems incomplete.

Author Response

Thank you very much for taking the time to review this manuscript. Please find the detailed responses below and the corresponding revisions highlighted changes in the re-submitted files.

Comments 1: I would like to suggest that authors provide some specifications or more detailed descriptions for the abbreviations used in the abstract;

Response 1:

Thank you for your suggestion. We provide the following specifications for the abbreviations in the abstract:

①  DHHC:Asp-His-His-Cys tetrapeptide motif 

②  PPT1:Palmitoyl protein thioesterase-1 

③  GNS561:Ezurpimtrostat

Comments 2: In my opinion, it would be nice to add chemical schemes for S-palmitoylation, O-palmitoylation, and N-palmitoylation reactions within an additional figure.

Response 2:

Thank you for your suggestion. We have added the chemical schemes for S-palmitoylation, O-palmitoylation, and N-palmitoylation reactions, which is designated as Figure 1 in Page 5.

Comments 3: - I think it would be helpful to the reader if the legend for the red spring representing palmitoylation will be included in the figure.

Response 3:

We sincerely thank the reviewer for careful reading. As suggested by the reviewer, we have added the legend for the red spring representing palmitoylation in Figure 2 (Page 10), Figure 3 (Page 12) and Figure 4 (Page 13).

5. Additional clarifications

We feel great thanks for your valuable feedback. In addition to the above improvement, we have further inspected and updated the literatures, ensuring the novelty of the article. Furthermore, we inspected and improved the language of the review, making it easier for the readers to understand. Last but not least, we further checked the other errors in our article including the typographical errors.

Reviewer 4 Report

Comments and Suggestions for Authors

This review seeks to explore and understand palmitoylation, a crucial cellular process, and its significant role in cancer cell development and death. Through this comprehensive review, the authors aim not only to highlight the transformative potential of studying palmitoylation in cancer treatment but also to deepen the scientific community's understanding of the molecular mechanisms at play in cancer.

The authors pose the following question: Could this be palmitoylation the key to unlocking a new era of precision oncology? I hope so, because it would open a new door in cancer treatment. Palmitoylation affects protein stability, protein-protein interactions, membrane localization, and signaling transduction, thereb regulating tumor survival and progression.

Palmitoylation enzymes or palmitoylated proteins are potential targets for tumor treatment. The review is well structured, with current bibliographic references that bring the reader up to date with the latest advances in this field of research.

English needs to be improved: on some occasions it is too colloquial and on other occasions, the phrases are bombastic and far from scientific communication, which must be simple, clear and concrete.

Comments on the Quality of English Language

English needs a deep revision, to make it more direct for the reader.

Author Response

Thank you very much for taking the time to review this manuscript. Please find the detailed responses below and the corresponding revisions highlighted changes in the re-submitted files.

Comments 1: The authors pose the following question: Could this be palmitoylation the key to unlocking a new era of precision oncology? I hope so, because it would open a new door in cancer treatment. Palmitoylation affects protein stability, protein-protein interactions, membrane localization, and signaling transduction, thereby regulating tumor survival and progression.

Response 1: Thank you for your professional review work on this problem. “Could this be palmitoylation the key to unlocking a new era of precision oncology?” After receiving the comments, we learn more about the precision oncology and read an authoritative literature: Global research trends on precision oncology: A systematic review, bibliometrics, and visualized study. Therefore, we believe that advancements in our molecular understanding and technological innovation hold the promise of pioneering new therapeutic pathways, potentially revolutionizing cancer metabolism and offering fresh strategies to inhibit tumor growth and its evasion tactics. Considering these future possibilities, it becomes clear that targeted modulation of palmitoylation pathways may well usher in a new era of precision oncology. Our views have been provided in section “Future Perspective” in Page 16.

Comments 2: Palmitoylation enzymes or palmitoylated proteins are potential targets for tumor treatment. The review is well structured, with current bibliographic references that bring the reader up to date with the latest advances in this field of research.

Response 2: Thank you for your suggestion. After receiving this comment, we read amounts of literatures and added another 28 articles related to our topic. Moreover, we have updated our references by replacing the majority of citations prior to 2010 with recent authoritative works on the subject to ensure the contemporary relevance of our literature review.

Comments 3: English needs to be improved: on some occasions it is too colloquial and on other occasions, the phrases are bombastic and far from scientific communication, which must be simple, clear and concrete.

Response 3: Thank you for your suggestion. We have tried our best to improve the manuscript. These changes will not influence the content and framework of the paper and we have highlighted these changes in the revised manuscript. Hopefully, the revised manuscript could be acceptable for you.

5. Additional clarifications

We feel great thanks for your valuable feedback. In addition to the above improvement, we provided a scheme for the process of palmitoylation, making it easier for the readers to understand. Also, we improve the section “Future Perspective” (Page 15), giving a more detailed and forward-looking perspective for readers who interested in palmitoylation. Last but not least, we further checked the other errors in our article including the typographical errors.

Reviewer 5 Report

Comments and Suggestions for Authors

The authors have discussed about the multifaceted role of palmitoylation across various cell death mechanisms in a comprehensive manner. Major points that the authors need to address are as follows:

1. The novelty of the article should be clearly highlighted as few reviews have already been published on this topic.
2. More important references from last few years should be discussed to improve visibility and quality of current work.
3. The search strategy used for the literature review should be indicated.
4. The quality and description of figures included should be improved.

5. The various limitations associated with drugs targeting palmitoylation should be discussed.
6. The authors should provide their own justification and relevance of the study. This will help the readers to understand the importance of the paper.

7. The manuscript should be carefully checked for typographical errors.

Comments on the Quality of English Language

Moderate editing required.

Author Response

Thank you very much for taking the time to review this manuscript. Please find the detailed responses below and the corresponding revisions highlighted changes in the re-submitted files.

Comments 1: The novelty of the article should be clearly highlighted as few reviews have already been published on this topic.

Response 1: Thank you for your suggestion. Our work spotlights the protein palmitoylation's emerging role in cell death regulation and oncogenesis. This emphasis has been incorporated at the conclusion of the "Introduction" section (Page 2), underscoring the pivotal relevance of our paper in this field.

Comments 2: More important references from last few years should be discussed to improve visibility and quality of current work.

Response 2: Thank you for your suggestion. After receiving this comment, we read amounts of literatures and added another 28 articles related to our topic. Moreover, we have updated our references by replacing the majority of citations prior to 2010 with recent authoritative works on the subject to ensure the contemporary relevance of our literature review. (Page 17-23)

Comments 3: The search strategy used for the literature review should be indicated.

Response 3: Thank you for your suggestion. We believe search strategy is very important for the readers to understand and re-do our work. Therefore, we have provided the details of how we search the literatures in the section “Methods”(Page 16).

Comments 4: The quality and description of figures included should be improved.

Response 4: Thank you for your suggestion. We have tried our best to improve the manuscript. These changes will not influence the content and framework of the paper. Hopefully, the revised manuscript could be acceptable for you.

Comments 5: The various limitations associated with drugs targeting palmitoylation should be discussed.

Response 5: Thank you for your valuable suggestion. We have indeed acknowledged and incorporated the limitations of the studies reported, specifically pertaining to GNS561, DC661, and erianin, within our manuscript. It is important to note that as the body of research on these compounds is relatively limited, there is a consequent lack of comprehensive data on their potential side effects (Page 13-14).

Comments 6: The authors should provide their own justification and relevance of the study. This will help the readers to understand the importance of the paper.

Response 6: Thank you for your valuable suggestion. We acknowledge the critical importance of establishing the relevance and significance of our study. As the current body of research scarcely addresses the role of palmitoylation in cell death, our work endeavors to bring to the forefront the significance of protein palmitoylation in the regulation of cell death and its implications in oncogenesis. We aim to furnish a comprehensive perspective that will aid readers in unraveling the intricate relationship between palmitoylation and cell death. To this end, we have underscored the importance of our study towards the last paragraph of the “Introduction” section (Page 2), elucidating how our findings could contribute to the existing knowledge and potentially guide future research in this domain.

Comments 7: The manuscript should be carefully checked for typographical errors.

Response 7: Thank you for your careful reading. We have corrected the typographical errors according to the template provided by Cancers.

4. Comments on the Quality of English Language: Moderate editing required.

Response: Thank you for your suggestion. We have tried our best to improve the language of the manuscript, which is highlighted in the revised manuscript. These changes will not influence the content and framework of the paper. Hopefully, the revised manuscript could be acceptable for you.

5. Additional clarifications

We feel great thanks for your valuable feedback. In addition to the above improvement, we provided a scheme for the process of palmitoylation, making it easier for the readers to understand. Also, we improve the section “Future Perspective” (Page 15), giving a more detailed and forward-looking perspective for readers who interested in palmitoylation. Last but not least, we further checked the other errors in our article including the typographical errors.